

# Combining data assimilation and machine learning to emulate a dynamical model from sparse and noisy observations: a case study with the Lorenz 96 model

Julien Brajard[1,2], Alberto Carrassi[1,3], Marc Bocquet[4], and Laurent Bertino[1]

[1]Nansen Center, Thormøhlensgate 47, 5006, Bergen, Norway
[2]Sorbonne University, CNRS-IRD-MNHN, LOCEAN, Paris, France
[3]Geophysical Institute, University of Bergen, Norway
[4]CEREA, joint laboratory École des Ponts ParisTech and EDF R&D, Université Paris-Est, Champs-sur-Marne, France

**Correspondence:** Julien Brajard (julien.brajard@locean-ipsl.upmc.fr)

**Abstract.**

A novel method, based on the combination of data assimilation and machine learning is introduced. The new hybrid approach is designed for a two-fold scope: (i) emulating a hidden, possibly chaotic, dynamics and (ii) predicting its future states. The method applies alternatively a data assimilation step, here an ensemble Kalman filter, and a neural network. Data assimilation is used to combine optimally a surrogate model with sparse noisy data. The resulting analysis is spatially complete and can thus be used as a training set by the neural network to upgrade the surrogate model. The two steps are then repeated iteratively. Numerical experiments have been carried out using the chaotic Lorenz 96, a 40-variables model, proving both convergence and statistical skills. The skill metrics include the short-term forecast skills out to two Lyapunov times, the retrieval of positive Lyapunov exponents and the power density spectrum. The sensitivity of the method to critical setup parameters is also presented: forecast skills decrease smoothly with increased observational noise but drops abruptly if less then half of the model domain is observed. The synergy demonstrated with a low-dimensional system is encouraging for more sophisticated dynamics and motivates further investigation to merge data assimilation and machine learning.

## 1 Introduction

Geophysical fluid dynamics is a domain of science in which the physical and dynamical laws governing the systems are reasonably well known. This knowledge is expressed by partial differential equations that are then discretised into numerical models (Randall et al., 2007). However, due to the misrepresentation of unresolved small-scales features or to neglected physical processes, parts of the numerical models have to be represented by empirical sub-models or parameterizations. Earth observations are thus needed for two tasks: first for model tuning and selection (best realism of the model; i.e. estimating/selecting the best possible parametrizations; see e.g. Metref et al., 2019) and then for data assimilation (best accuracy of the model;





i.e. estimating the system state; see e.g. Carrassi et al., 2018). Sometimes the same data are used for both tasks. In the last decades, the volume and quality of these observations have increased dramatically, particularly thanks to remote sensing (see, e.g., Kuenzer et al., 2014).

At the same time, new developments in machine learning, particularly deep learning (Lecun et al., 2015), have demon-
strated impressive skills in reproducing complex spatiotemporal processes (see e.g. Tran et al., 2015, for video processing) by efficiently using a huge amount of data, thus paving the path for their use in Earth system science (Reichstein et al., 2019).

Forecasting and simulating are two different objectives but can both be achieved by using machine learning (ML). Various ML algorithms have been already applied to produce surrogate models of fully observed low-order chaotic systems and then used for forecasting purposes; examples include reservoir modeling (Pathak et al., 2017, 2018), residual neural network (Fa-
blet et al., 2018) and recurrent networks (Park and Yan Zhu, 2002; Park, 2010). Machine learning has also been applied for nowcasting based on real observations, such as sea surface temperature (de Bezenac et al., 2017) or precipitations  (Shi et al., 2015, 2017). Along with these advancements in ML, the issue of determining a surrogate model of an unknown underlying process based on observations has also been addressed using sparse regression (Brunton et al., 2017) and, more recently, data assimilation (Bocquet et al., 2019).

Examples of numerical results have proven the effectiveness of ML to reconstruct dynamics under the condition that the ML algorithm is trained on noise-free and complete observations of the system (i.e. when the system state is fully observed). Nevertheless, these circumstances are almost never found in practice and certainly not in the geosciences. In the case of partial and noisy observations, one can use a nearest-neighbour or an analogue approach consisting in finding similar past data (if available) as a forecast (Lguensat et al., 2018). Machine learning techniques were also applied where only a dense portion of
the system is observed (e.g. de Bezenac et al., 2017; Lu et al., 2017) or when the observations are sub-sampled in time (Nguyen et al., 2019).

Most of the ML algorithms used in the aforementioned studies are not suited to realistic cases of noisy and sparse observations. By "sparse", we mean here that the state of the system is not densely observed, and even the locations and the number of the observation may vary in space and time.

Data assimilation (DA) is a natural framework to deal with this observational scenario. Data assimilation aims at estimating the state of a system given noisy and unevenly distributed observations and a dynamical model (Carrassi et al., 2018). The output of DA depends explicitly on uncertainties in the numerical model (Harlim, 2017), which has led to developing techniques for taking into account model errors in the DA process. This can be done by a parametrization of the model error embedded in the model equations (Aster et al., 2005; Carrassi and Vannitsem, 2011; Bocquet, 2012) or in form of a stochastic noise
added to the deterministic model (e.g. Trémolet, 2006; Ruiz et al., 2013; Raanes et al., 2015; Sakov et al., 2018). In any case, a dynamical model must be a priori provided. In Bocquet et al. (2019) though, this constraint is relaxed to the point where only the general form of the differential equations is specified (e.g. linear and quadratic): it was shown that the optimization problem solved by DA in presence of a model error is equivalent to a ML problem. A similar equivalence was also shown by Abarbanel et al. (2018), but starting from the point of view of a ML problem. A pure DA approach however does not
leverage on recent ML developments, that bring flexibility and facilitate parallel calculations. By including explicit or implicit





regularization processes, ML algorithms also enable the possibility of optimization in high-dimension without the need for additional information under the form of an explicit prior.

This paper stands at the crossroads of DA and ML in cases where both the system state and its dynamical model have to be put out based on noisy and sparse observations. The proposed hybrid algorithm relies on DA to estimate the state of the system

and on ML to emulate the surrogate model.

In section 2, we give a formulation of the problem and detail how DA and ML are combined. In section 3, we present the experimental setup using the 40-variables Lorenz model (Lorenz and Emanuel, 1998) chosen to illustrate the approach. Section 4 describes the numerical results using different metrics and discusses the algorithm sensitivity to the number of observations and their noise statistics as well as to control parameters. Conclusions and perspective are drawn in section 5.

## 2 Methodology

### 2.1 Definition of the problem

Let us consider a time series of multi-dimensional observations $\mathbf{y}_k^{\mathrm{obs}} \in \mathbb{R}^p$ of an unknown process $\mathbf{x}_k \in \mathbb{R}^m$:

$$\mathbf{y}_k^{\mathrm{obs}} = \mathcal{H}_k(\mathbf{x}_k) + \boldsymbol{\epsilon}_k^{\mathrm{obs}}, \tag{1}$$

where $0 \leq k \leq K$ is the index corresponding to the observation time $t_k$, and $\mathcal{H}_k : \mathbb{R}^m \to \mathbb{R}^p$ is the observation operator (sup-

posed to be known). The observation error $\boldsymbol{\epsilon}_k^{\mathrm{obs}}$ is assumed to follow a normal distribution of mean zero and covariance matrix $\mathbf{R}_k$.

In the following we will assume for the sake of simplicity that the number of observations, $p$, and their noise level do not change with time. Furthermore, we will assume the observations to be spatially uncorrelated, implying that $\mathbf{R}$ is diagonal such as: $\mathbf{R}_k \equiv (\sigma^{\mathrm{obs}})^2 \mathbf{I}_p$ where $\mathbf{I}_p$ is the identity matrix of size $p$. We will also consider a regular time discretisation step such as:

$t_{k+1} - t_k = h$ for all $k$.

We suppose that $\mathbf{x}_k$ is a time-discretisation of a continuous process $\mathbf{x}$, which obeys to an unknown ordinary differential equation of the form

$$\frac{\mathrm{d}\mathbf{x}}{\mathrm{d}t} = \mathcal{M}(\mathbf{x}). \tag{2}$$

Our goal is to derive a surrogate model $\mathcal{G}$ of the resolvent of $\mathcal{M}$ between $t_k$ and $t_{k+1}$ (hereafter called in short: the surrogate

model):

$$\mathbf{x}_{k+1} = \mathcal{G}(\mathbf{x}_k) + \boldsymbol{\epsilon}_k^{\mathrm{m}} = \mathbf{x}_k + \int_{t_k}^{t_{k+1}} \mathcal{M}(\mathbf{x}) \, \mathrm{d}t, \tag{3}$$

where $\boldsymbol{\epsilon}_k^{\mathrm{m}}$ is the error of the surrogate model $\mathcal{G}$.



## 2.2 Convolutional neural network as surrogate model

As stated in the introduction, several papers have used convolutional neural networks for representing surrogate models (see, e.g., Shi et al., 2015; Fablet et al., 2018; de Bezenac et al., 2017). Equation (3) being in the incremental form $\mathbf{x}_{k+1} = \mathbf{x}_k + \cdots$, one-block residual networks are suitable (He et al., 2016). So, our neural network can be expressed as a parametric function $\mathcal{G}_{\mathbf{W}}(\mathbf{x})$:

$$\mathcal{G}_{\mathbf{W}}(\mathbf{x}_k) = \mathbf{x}_k + f_{\mathrm{nn}}(\mathbf{x}_k, \mathbf{W}), \tag{4}$$

where $f_{\mathrm{nn}}$ is a neural network and $\mathbf{W}$ its weights; $f_{\mathrm{nn}}$ is composed of convolutive layers (Goodfellow et al., 2016). Convolutive layers apply a convolution acting locally around each grid point of the field. It is equivalent to a locality hypothesis, assuming that there are no long-range correlations between the state variables. Note that it does not discard further distance correlation arising from the time integration.

The determination of the optimal weights $\mathbf{W}$ is achieved via an iterative minimization process of a loss function: this process is referred to as the *training phase*. The loss function reads:

$$L(\mathbf{W}) = \sum_{k=0}^{K-N_{\mathrm{f}}-1} \sum_{i=1}^{N_{\mathrm{f}}} \left\| \mathcal{G}_{\mathbf{W}}^{(i)}(\mathbf{x}_k) - \mathbf{x}_{k+i} \right\|_{\mathbf{P}_k^{-1}}^2, \tag{5}$$

where $N_{\mathrm{f}}$ is the number of time steps corresponding to the forecast lead time for which the error between the simulation and the target is minimized; $\mathbf{P}_k$ is a symmetric, semi-definite positive matrix defining the norm $\|\mathbf{x}\|_{\mathbf{P}_k^{-1}}^2 = \mathbf{x}^{\mathrm{T}} \mathbf{P}_k^{-1} \mathbf{x}$. Note that $\mathbf{P}_k$ plays the role of the surrogate model error covariance matrix. We will clarify in the next sections how $\mathbf{P}_k$ is itself estimated using DA.

The formalism used here is close to what is proposed in Fablet et al. (2018); E (2017); Chang et al. (2017) in which the neural network is identified with a dynamical model. One major difference is that we do not aim here at identifying the dynamical model, but reproducing a resolvent of the underlying dynamics. We also highlight that a prerequisite to train this neural network is to have access to the time series of the complete state field $\mathbf{x}_{1:K}$ for which we will rely upon DA.

## 2.3 Data assimilation

In this work, we use the finite-size ensemble Kalman filter (Bocquet, 2011; Bocquet et al., 2015), hereafter denoted EnKF-N. However, our approach is not tied to any particular DA algorithm and is straightforwardly extendable to any adequate DA method for the problem at hand. For example, a smoother would lead to more accurate results but would be more costly and is thus less common in the operational DA community. Our choice of the EnKF-N is motivated by its efficiency, its high accuracy for low-dimensional systems, and its implicit estimation of the inflation that would otherwise have had to be tuned.

The EnKF-N is a sequential ensemble DA technique. The analysis (resp. forecast) matrix at time $t_k$ is defined by $\mathbf{X}_k^{\mathrm{a/f}} \equiv \left[ \mathbf{x}_{k,1}^{\mathrm{a/f}}, \cdots, \mathbf{x}_{k,p}^{\mathrm{a/f}}, \cdots, \mathbf{x}_{k,N}^{\mathrm{a/f}} \right] \in \mathbb{R}^{m \times N}$, where the subscript a (resp. f) stands for the analysis (resp. forecast). The members $\mathbf{x}_{k,p}^{\mathrm{f}}$ are defined recursively in the forecast step:

$$\mathbf{x}_{k,p}^{\mathrm{f}} = \mathcal{G}(\mathbf{x}_{k-1,p}^{\mathrm{a}}) + \boldsymbol{\epsilon}_{k,p}^{\mathrm{m}}, \tag{6}$$



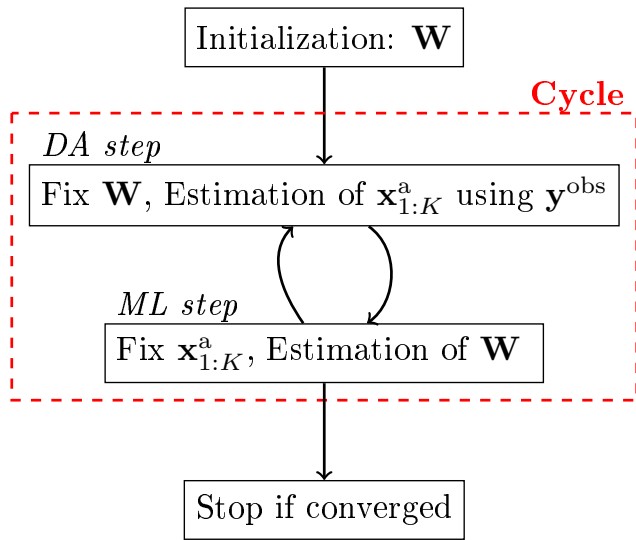

**Figure 1.** Scheme of the algorithm. The two-step procedure DA (data assimilation) followed by ML (machine learning) constitutes one cycle of the algorithm. $\mathbf{P}_k$ is passed from the DA to the ML step.

where

$\epsilon_{k,p}^{\mathrm{m}}$ is the model noise for member $p$ at time $t_k$.

In the analysis step, given the observations $\mathbf{y}_k^{\mathrm{obs}}$ and the forecast ensemble $\mathbf{X}_k^{\mathrm{f}}$, we derive an updated ensemble $\mathbf{X}_k^{\mathrm{a}}$ following the algorithm 3.4 in Bocquet (2011). We estimate the mean of the ensemble state $\mathbf{x}_k^{\mathrm{a}}$ and the error covariance matrix $\mathbf{P}_k^{\mathrm{a}}$:

$$\mathbf{x}_k^{\mathrm{a}} = \frac{1}{N}\sum_{p=1}^{N}\mathbf{x}_{k,p}^{\mathrm{a}}, \tag{7}$$

$$\mathbf{P}_k^{\mathrm{a}} = \frac{1}{N-1}(\mathbf{X}_k^{\mathrm{a}} - \mathbf{x}_k^{\mathrm{a}}\mathbf{1}_N)(\mathbf{X}_k^{\mathrm{a}} - \mathbf{x}_k^{\mathrm{a}}\mathbf{1}_N)^{\mathrm{T}}, \tag{8}$$

where $\mathbf{1}_N$ is the row vector composed of N ones.

## 2.4 Combining data assimilation and machine learning

The general idea of the proposed algorithm is to combine DA and ML: the neural network provides a surrogate forward model to DA and, reciprocally, DA provides a time series of complete states to train the neural network. An illustration of the algorithm is displayed in Fig. 1. At each cycle, a DA step feeds into a ML step and vice-versa. The DA step is applied to the whole observation time series according to Eq. (6) and Eq. (7) using the surrogate model $\mathcal{G}_W$ as the forecast model. The ML step is done by minimizing the loss in Eq. (5). In this work, the matrix $\mathbf{P}_k$ used in the norm of the loss $L(\mathbf{W})$ is the diagonal of the matrix defined in Eq. (8). This simplification is done to gain computational efficiency and to avoid numerical bias potentially arising from biased estimation of the covariance. However besides this simplification it is worth noticing that, being based on the ensemble, the covariance matrix $\mathbf{P}_k^{\mathrm{a}}$ is time-dependent like the analysis error in the state estimate.





The procedure can be viewed as an expectation-maximization algorithm in which the DA is the expectation step and ML is the maximization step (Ghahramani and Roweis, 1999; McLachlan and Krishnan, 2007; Nguyen et al., 2019). Two particular features of the proposed algorithm can be highlighted here:

– We leverage on ML algorithms and libraries to estimate the parameters of the surrogate model $\mathcal{G}_W$. We can thus benefit
from the high speedups, parallel computing and regularization processes of these libraries enabling optimization of high-dimensional $\mathbf{W}$.

– The two steps in each cycle are separate, making the algorithm very flexible. The choice of ML and DA algorithm are independent. They can be changed or combined with an external system.

The initialization step is critical as the convergence of the hybrid algorithm is not guaranteed for any initial choice of $\mathbf{W}$. It
is possible to initialize the weights of the neural network randomly, or to use an interpolated field to train the initial weights of the neural network.

We also found that the convergence of the algorithm is sensitive to hyper-parameters such as the model noise level, Eq. (6), or the forecast lead time $N_\mathrm{f}$, Eq. (5). These aspects are discussed in section 4.5.

## 3   Numerical experiment setup

### 3.1   Model Setup

Our combined DA-ML method is tested using the 40-variables Lorenz model (Lorenz and Emanuel, 1998) hereafter denoted L96 which is used to produce the synthetic observations. In this idealized case, the surrogate model can be compared with the real underlying dynamic, called the "true" model. The model L96 is defined on a periodic one-dimensional domain by the following set of ordinary differential equations:

$$\frac{\mathrm{d}x_n}{\mathrm{d}t} = (x_{n+1} - x_{n-2})x_{n-1} - x_n + F, \tag{9}$$

where $x_n$ ($0 \leq n < m$) is the scalar state variable, $x_m = x_0$, $x_{-1} = x_{m-1}$, $x_{-2} = x_{m-2}$, $m = 40$ and $F = 8$. The model is integrated using a fourth order Runge-Kutta scheme with a time step $h = 0.05$. The resolvent of the true model is denoted $\mathcal{G}_{L96}$. The model L96 with the current choice for $m$ and the forcing $F$ is chaotic, with the largest Lyapunov exponent $\Lambda_1 \approx 1.67$. In the following, we will use the Lyapunov time unit $t_\Lambda = \Lambda_1 t$ where $t$ is the time in model unit: one Lyapunov time unit
corresponds to the time for the error to grow by a factor $e$.

Unless stated otherwise, results will be shown on the so-called "reference setup" specified in the following. The model is integrated over $40,000$ time steps ($K = 40,000$) to produce a state vector time series $\mathbf{x}_{1:K}$ considered as the truth. In this setup, the observation operator $\mathcal{H}_k$ is defined as a sub-sampling operator that draws randomly $p = 20$ values at each time step (corresponding to 50% of the field) from a uniform distribution changing the observation locations at each time step. The
observation interval is the same as the integration time step of the model. In a real application, the integration time of the





underlying model would have been unknown (the underlying model might not even be specified), so the only known time step is that of the observations (see the discussion in Bocquet et al., 2019). The standard deviation of the observational error in Eq. (1) is $\sigma^{\mathrm{obs}} = 1$ (about $5\%$ of the total range).

## 3.2 Scores

Given the resolvent of the true model $\mathcal{G}_{L96}$ and the true field $\mathbf{x}_{1:K}$, we compute the following scores:

– RMSE-a: aims at assessing the accuracy of the DA step at the end of the procedure using the spatiotemporal root mean square error (RMSE) of the analysis:

$$\text{RMSE-a} = \sqrt{\frac{1}{m(K - k_0)} \sum_{k=k_0}^{K} \sum_{n=0}^{m-1} (x_{n,k} - x_{n,k}^{\mathrm{a}})^2}, \tag{10}$$

where $\mathbf{x}_k = [x_{0,k}, \cdots, x_{m-1,k}]$ and $k_0 > 0$ is the first value considered in the evaluation (we have chosen $k_0 = 100$
corresponding to $t_\Lambda \approx 8$) so as to remove the effect of the initial state.

– RMSE-f: evaluates the forecast quality of the surrogate model $\mathcal{G}_{\mathbf{W}}$. We defined a vector of $P = 500$ different initial conditions $\mathbf{x}_0^{1:P}$ that are not consecutive in time and are drawn from another simulation of the true model. RMSE-f is calculated for a predicting horizon of $t_i - t_0$:

$$\text{RMSE-f}(t_i - t_0) =$$

$$\sqrt{\frac{1}{mP} \sum_{p=1}^{P} \sum_{n=0}^{m-1} \left([\mathcal{G}_{L96}^{(i)}(\mathbf{x}_0^p)]_n - [\mathcal{G}_{\mathbf{W}}^{(i)}(\mathbf{x}_0^p)]_n\right)^2} \tag{11}$$

– Lyapunov spectrum: assesses the long-term dynamical property of the surrogate model to be compared with that of the true model. The Lyapunov spectrum was computed experimentally over an integration of 100,000 time steps to ensure the convergence of the calculation (Wolf et al., 1985).

– Power spectral density (PSD): describes the "climate" of the model in terms of its frequency content. The PSD is com-
puted on a single grid point using the Welch method (Welch, 1967): the time series is split into 62 overlapping segments of 512 points. A power spectrum is computed on each segment convoluted with a Hanning window and all spectra are then averaged to yield a smooth PSD.

## 3.3 Surrogate model optimization setup

### 3.3.1 Data assimilation setup

Following results in Bocquet (2011), we work with an ensemble of size $N = 30$ which makes localization unnecessary. Note however that localization is unavoidable when working on high dimension and needs careful tuning (see, e.g., section 4.4.1 in





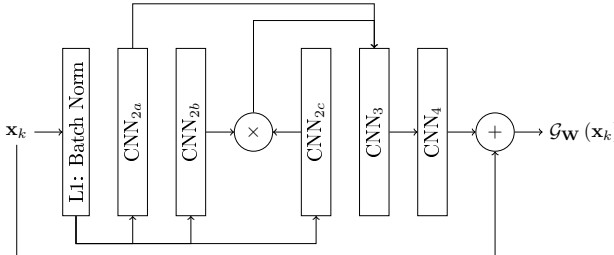

**Figure 2.** Proposed neural network architecture for the surrogate model. The input layer is to the left.

Carrassi et al., 2018). The initial state is drawn from a sequence calculated from a previous run. The model noise $\epsilon^{\mathrm{m}}$ is drawn from a normal distribution with zeros mean and standard deviation $\sigma^{\mathrm{m}} = 1$. The sensitivity to the level of model noise in the DA step is discussed in section 4.5.

### 3.3.2 Neural network setup

The architecture of the neural network (number/type of layers, activation functions, among other factors) has been determined by cross-validation experiments on two datasets: one with a fully observed field without noise, the other with a quadratic interpolation of the observations from the reference setup ($p = 20$ observations, and $\sigma^{\mathrm{obs}} = 1$). The proposed architecture is shown in Fig. 2 whereas its parameters and those of the training phase are detailed in Tab. 1. This architecture is similar to the one proposed by Fablet et al. (2018), but given that our goal is not the identification of the model itself, our architecture is not

embedded in a time-integration step (here a fourth order Runge-Kutta scheme). In this way, the neural network is emulating the resolvent of the model but may not represent the model itself.

     The batch-norm (Ioffe and Szegedy, 2015) is used only for the input layer, so its main effect is to standardize the input data. The other layers are convolutional layers (CNN) as described in Goodfellow et al. (2016). The second bi-linear layer is composed of three sub-layers: the outputs of the sub-layers $\mathrm{CNN}_{2b}$ and $\mathrm{CNN}_{2c}$ are multiplied, the product obtained is then

added to the sub-layer $\mathrm{CNN}_{2a}$ to compute the output of the second layer. The third and fourth layers are standard CNNs. The neural network being residual, the output of the fourth layer is added to the input $\mathbf{x}_k$ to compute the output of the neural network. The number of epochs (20) used for the training is the number of times each element in the training dataset is used by the neural network in each cycle for optimizing the neural network's weights. Therefore if, for example, the algorithm carries out 50 cycles, the neural network performs a total of 1000 optimization epochs ($20 \times 50$) after the initialization step.

### 3.3.3 Initialization of the weights

The weights of the neural network must be initialized before the first cycle (see Fig. 1). This is done by training the neural network on a dataset produced by a quadratic interpolation (in space and time) of the observations. To account for the lower quality of interpolated data, values of $\mathbf{P}_k^{-1}$ in the loss function in Eq. (5), are set to 1 at the locations of observations and zero where they were interpolated. By doing so, interpolated values of missing data are only used as an input of the neural network



**Table 1.** Setup of the architecture (cf Fig. 2) and the training phase of the surrogate neural network model.

| Neural Network Architecture | |
| --- | --- |
| Input size | 40 |
| Output size | 40 |
| Number of layers | 4 |
| Number of weights | 9389 |
| Type of layer 1 | batch-norm[a] |
| Type of layer 2 | bi-linear Convolutive |
| Size of layer 2 | $24 \times 3 = 72$ |
| Layer 2 convolutive kernel size | 5 |
| Activation function | Rectifier linear unit |
| Type of layer 3 | Convolutive |
| Size of layer 3 | 37 |
| Layer 3 convolutive kernel size | 5 |
| Activation function | Rectifier linear unit |
| Type of layer 4 | Convolutive |
| Size of layer 4 | 1 |
| Layer 4 convolutive kernel size | 1 |
| Layer 4 regularization | $L2(10^{-4})$ |
| Activation function | linear |
| Training | |
| Optimizer | Adagrad[b] |
| mini-batch size | 256 |
| number of epochs | 20 |
| Forecast lead time $N_{\mathrm{f}}$ | 1 |

[a] Ioffe and Szegedy (2015)
[b] Duchi et al. (2011)

but not as a target. After cross-validation, it has been chosen to set initially $N_{\mathrm{f}} = 4$ and the number of epochs to 40 for this first training. These particular high values can be explained by the fact there is no prior knowledge of the initial weights for the first training, and thus the initial neural network is far from producing a realistic surrogate model. Consequently, more epochs and more target values in the loss function are needed to make the neural network to converge toward a first surrogate model. The

5    other parameters of the training phase are given as in Tab. 1.



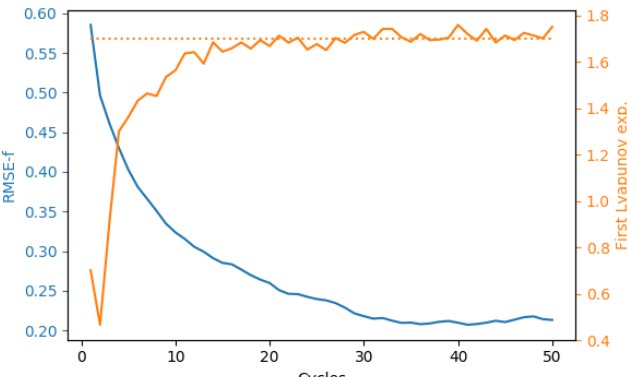

**Figure 3.** RMSE-f and first Lyapunov exponent as a function of the algorithm cycle. Half of the state-vector is observed, at randomly varying locations. Observational noise standard deviation is $\sigma^{\mathrm{obs}} = 1$. The dotted line corresponds to the first Lyapunov exponent of the true model (=1.67)

## 4 Results and discussion

### 4.1 Convergence of the algorithm

Figure 3 shows the evolution of the RMSE-f and of the first Lyapunov exponent after successive training cycles. The RMSE-f decreases to 0.21 while the first Lyapunov exponent approaches 1.67, which is the target value of the true L96 dynamics. It can
5  be noticed that none of these quantities are explicitly optimized within a cycle. This suggests that the proposed algorithm is converging regarding two very different targets: RMSE-f is a forecast skill (a sign of accuracy) and the first Lyapunov exponent is a property of the long-term dynamics (a sign of consistency).

Furthermore, the high value of the Lyapunov exponent obtained from the first cycle may be due to the different parameters $N_{\mathrm{f}}$ and $\mathbf{P}_k$ used in the loss function for the initial training of the neural network (see section 3.3.3).

10  Fig. 4 shows the PSD of the true L96 model together with the surrogate model after the first cycle (grey line) and after convergence (orange line). After 1 cycle, some frequencies are favoured (see the peak at $\sim 0.8$Hz) and indicate that the periodic signals are learnt first. Results after convergence are discussed in section 4.3.

Given that our primary goal here is to demonstrate the accuracy and convergence of the method rather than optimizing the computational cost, we have intentionally waited for a large number of iterations (50) before stopping the algorithm and
15  retained the surrogate model corresponding to the lowest RMSE-f. A more sophisticated stopping criterion would have reduced the computational burden and would be required in applications in high dimension.

### 4.2 Interpolation

The observations being a sub-sample from the full state vector implies that DA acts as an interpolator and a smoother of the observations, producing a so-called "analysis". Regarding the complete algorithm merging DA and ML, the interpolation of





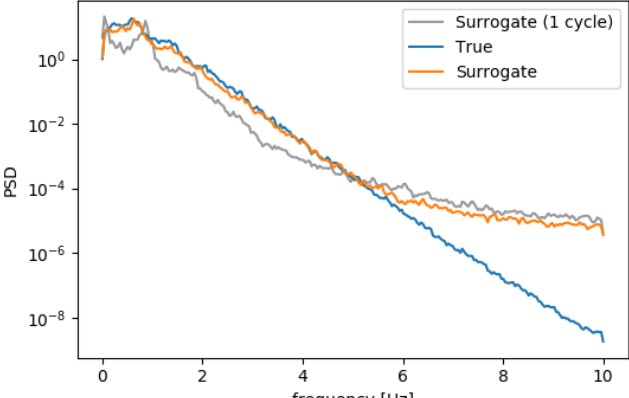

**Figure 4.** Power density spectrum of the surrogate model after one cycle (in gray) and after convergence (in orange) compared with the true model (in blue). 50% of the field was observed with a noise $\sigma^{\mathrm{obs}} = 1$.

**Table 2.** RMSE-a of the complete field estimated with 50% of observation and a noise $\sigma^{\mathrm{obs}} = 1$

| Method | RMSE-a |
|---|---|
| DA with true model | 0.34 |
| DA with surrogate model | 0.80 |
| Quadratic interpolation | 2.32 |

the observations can be viewed as a side-product. To evaluate the analysis, we compare the RMSE-a with two extreme upper and lower bounds. The former is the RMSE of a field obtained via quadratic interpolation without any use of a dynamical model (which is instead essential in DA); the latter is the RMSE of the analysis obtained using the EnKF-N with the true L96 model instead of the surrogate.

5     Results are reported in Tab. 2. It can be seen that the use of the surrogate model as an ingredient in the DA leads to a significant error reduction over the quadratic interpolation, by a factor of 3 (cf the second and third line in Tab. 2). Furthermore, the RMSE-a is below the observation standard deviation (0.8; recall that $\sigma^{\mathrm{obs}} = 1$) and testifies of the healthy functioning of DA. On the other hand, the surrogate model is less accurate than the true model by a factor of 2 (cf the first and second rows in Tab. 2). This means that the data-driven surrogate model does not carry all the dynamics of the underlying process, which

10   should be expected given that the observations are partial and noisy. Recall also that the true model output is hidden from the neural network.

    These results suggest that this method could also be used to interpolate spatiotemporal fields in the absence of assumption on the underlying dynamics. More generally, they indicate that the success of the method is coming both from DA and ML, and not from one of the two approaches alone.



### 4.3 Emulating the underlying dynamics

The capabilities of our approach to emulate the underlying dynamics which has generated the observations is assessed by comparing the PSD and the Lyapunov spectrum of the surrogate model with those of the true model.

The PSD is shown in Fig. 4. The result of the first cycle has already been discussed in section 4.1. After convergence, it can
be noticed that the surrogate model reproduces the spectrum up to 5 Hz (0.2 time units corresponding to 0.33 Lyapunov time units) but then adds high-frequency noise.

Note that the PSD has been computed using a long simulation (1336 Lyapunov time units, corresponding to 16,000 time steps), which means that the surrogate model is stable enough to compute long-term simulations.

Figure 5 shows the 40 Lyapunov exponents of both models. Remarkably, the positive part of the spectrum (the first twelve
exponents) of both models are very close to each other. This indicates that the surrogate model possesses an unstable subspace (the time-dependent space where perturbations grow) with the same general properties, size and average growth rate, as the truth. This means that initial errors would grow, on average, in a similar way in the two models that will also share the same e-folding time (uniquely determined by the first Lyapunov exponent) and Sinai-Kolmogorov entropy (given by the sum of the positive Lyapunov exponents). The situation is different in the null and negative parts of the spectrum, where the surrogate
model possesses smaller (larger in absolute value) exponents. A closer inspection reveals that this shift is mainly due to the difference in the null part of the spectrum. Recall that the L96 possesses one null exponent and two very small ones; the surrogate model, on the other hand, displays a single null exponent. The difficulty to reproduce the null part of the Lyapunov spectrum has been also put forward by Pathak et al. (2017). While a robust explanation is still missing we argue here that this behaviour can be related to the known slower convergence of the instantaneous null exponent(s) toward their asymptotic
values compared to the positive and negative parts of the spectrum (linear versus exponential; see Bocquet et al., 2017). It is finally worth noting that the average flow divergence, which is given by the sum of the Lyapunov exponents, is smaller in the surrogate model than in the truth, implying that volumes in its phase space will contract, on average, faster than in the case of the true model (volumes are bound to contract given the dissipative character of the dynamics). The average flow divergence also drives the evolution of the probability density function (PDF) of the state in the phase space according to the Liouville
equation: the results in Fig. 5 thus suggest that, once the PDF is interpreted as an error PDF about the system state estimate, the surrogate model is, on average, slightly more predictable in the sense that the PDF support shrinks faster.

### 4.4 Forecast skills

In Fig. 6, we compare one simulation of the true model (top panel) with the surrogate model (middle panel) for about 8 Lyapunov time units. Both simulations have the same initial condition. It can be noticed that the difference (bottom panel)
between the true and the surrogate simulation is increasing with respect to time. After 2 Lyapunov time units, the trajectory of the surrogate model diverges significantly from that of the true model and the error saturates after 4-5 Lyapunov time units.

We quantify the effect of noise on the forecast skill in Fig. 7 which shows the RMSE-f of the surrogate model as a function of the forecast lead time (panel (a)) and of the observational noise level (panel (b)) in the case of a 50% observed field. As



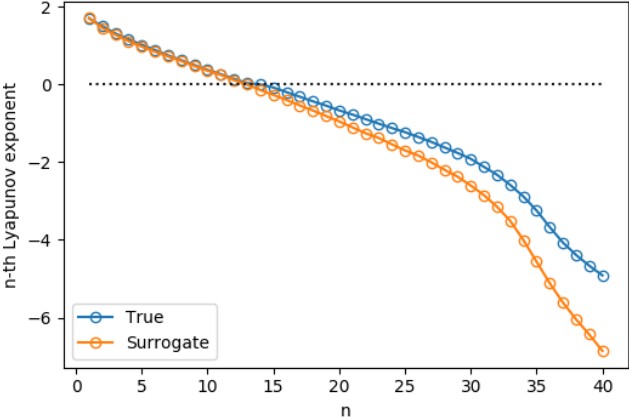

**Figure 5.** 40 Lyapunov exponents of the surrogate model (in orange) compared with the true model (in blue). 50% of the field was observed with a noise $\sigma^{\mathrm{obs}} = 1$.

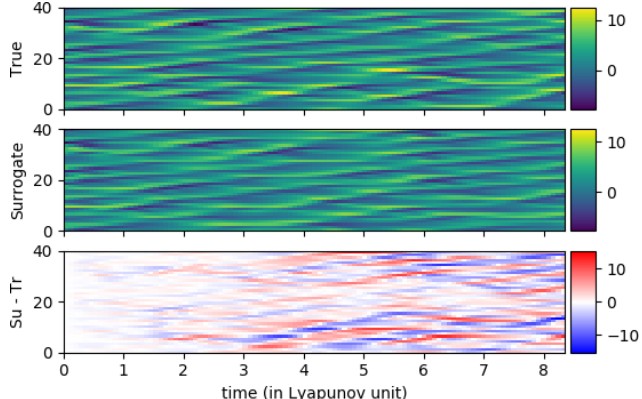

**Figure 6.** Hovmøller plot of one trajectory of the true model, the surrogate model and their difference given the same initial condition with respect with the lead time (in Lyapunov time). 50% of the field is observed with a noise level $\sigma^{\mathrm{obs}} = 1$.

expected, the forecast skills deteriorate gradually as the observation noise increases. It is worth noticing that the asymptotic values of RMSE-f are inversely proportional to the noise level. This is due to the fact that the surrogate model trained with noisy observations underestimates the variance of the forecast (the spatial variability is reduced). It can also be qualitatively seen in Fig. 6 in which it appears that the surrogate model produces less extreme values than the true model.

5  In Fig. 8, we study the sensitivity of the forecast skill to the observation density. Observational noise is fixed to $\sigma^{\mathrm{obs}} = 1$ following the protocol detailed in section 3.1 but the fraction of the observed model domain varies from 30% to 100%. The RMSE-f of the surrogate model trained with complete (i.e. 100%) perfect (i.e. $\sigma^{\mathrm{obs}} = 0$) observations is also displayed for reference. Since the observations are "perfect" no DA has been carried out. The RMSE-f at $t_0 + h$ (see right-most bar in panel (b)) is $0.014$ which is close to the value in Fablet et al. (2018) despite the fact that, in our case, the model is not identifiable. If





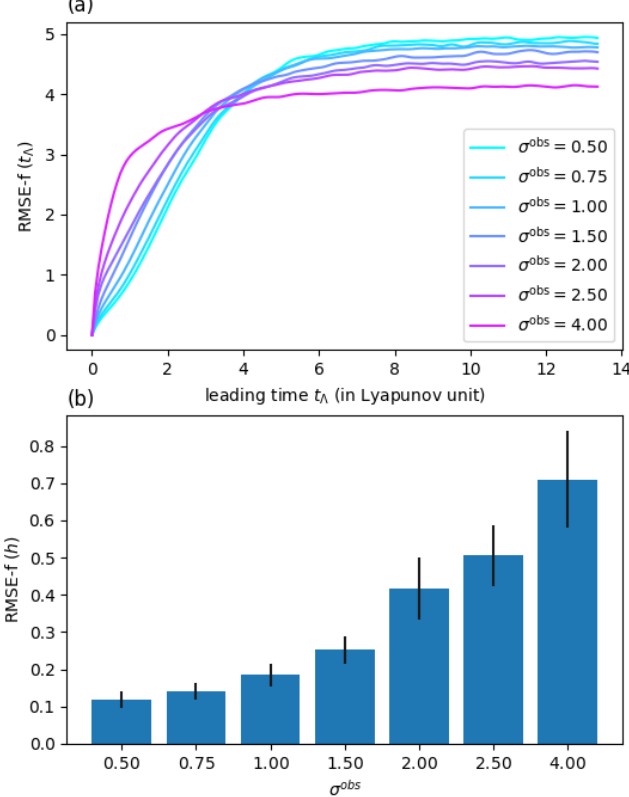

**Figure 7.** (a): RMSE-f of the surrogate model compared with the true model with respect to the leading time for different observational error standard deviation. (b): RMSE-f for the first time step of the surrogate model with respect to the level of noise, vertical black lines represent the standard deviation of the RMSE-f. The surrogate model has been trained on different noise levels ($\sigma^{obs}$ from 0 to 4). The values are averaged over simulations starting from 500 different initial conditions at $t = 0$. 50% of the field is observed.

the field is observed with less than 50% coverage, the skills are significantly degraded. But if more than 50% of observations are available, there is no obvious improvement in terms of RMSE-f. These results suggest that, to a certain point, the DA phase of the algorithm is very efficient in producing accurate analyses from a sparsely observed field, which is also confirmed by the scores in Tab. 2. The threshold of 50% of observations is likely to be a value specific to the L96 model. Note also that we have used the same set of parameters for all the experiments presented (architecture of the neural net, additive stochastic model noise, ...) whereas specific tuning for each experimental setup would be desirable and could lead to improved performances.

## 4.5 Sensitivity to hyper parameters

We study here the effect of the forecast lead time $N_f$ included in the loss, as defined in Eq. (5) and of the model noise by varying its standard deviation $\sigma^m$ defined in Eq. (6). In principle, these parameters should have been tuned for each cycle and observational time series considered. Nevertheless, we chose a common tuning to facilitate the comparison among cases. In



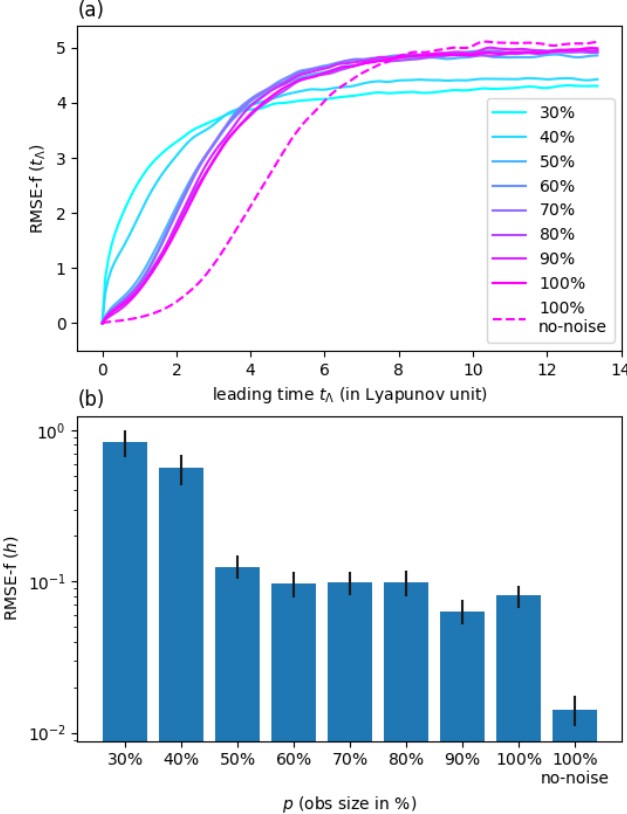

**Figure 8.** (a): RMSE-f of the surrogate model compared with the true model with with respect to the leading time for different observation densities. (b): RMSE-f for the first time step of the surrogate model with respect to the observation density, vertical black lines represent the standard deviation of the RMSE-f. The surrogate model has been trained with increasing observation density (from 30% to 100%). The values are averaged over simulations starting from 500 different initial conditions. The noise of the observation is $\sigma^{\text{obs}} = 1$ except for the "100% no-noise" value that corresponds to $\sigma^{\text{obs}} = 0$

.

the following, we evaluate the effect of these two parameters on the reference setup ($p = 20$ and $\sigma^{\text{obs}} = 1$) using two metrics: the RMSE-f defined in Eq. (11) and the RMSE between the first 12 exponents of the Lyapunov spectrum (corresponding to the positive Lyapunov exponents in the true model):

$$\text{RMSE-Lyapunov} = \sqrt{\sum_{n=1}^{12} \left( \Lambda_n^{\mathcal{G}_{\text{L96}}} - \Lambda_n^{\mathcal{G}\mathbf{w}} \right)^2}, \tag{12}$$

5    where $\Lambda_n^{\mathcal{G}_{\text{L96}}}$ (resp. $\Lambda_n^{\mathcal{G}\mathbf{w}}$) is the $n$-th Lyapunov exponent for the true model (resp. surrogate model). While the RMSE-f estimates the forecast skills of the surrogate model, the RMSE-Lyapunov assesses the algorithm capability to reconstruct the long-term chaotic dynamics.





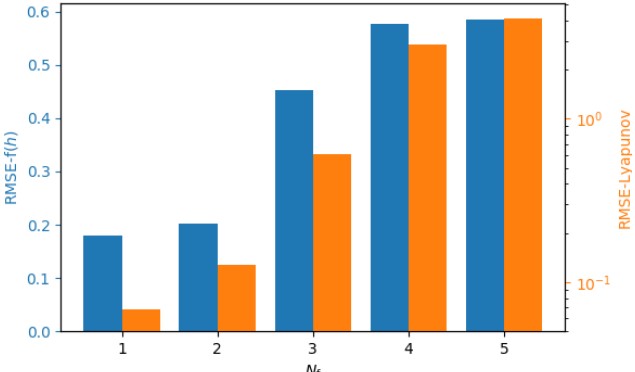

**Figure 9.** RMSE-f (blue) and RMSE-Lyapunov (orange) with respect to the forecast lead time $N_\mathrm{f}$, Eq. (5). 50% of the field was observed with a noise $\sigma^\mathrm{obs} = 1$.

### 4.5.1 Target lead time for training

Figure 9 represents the value of RMSE-f and RMSE-Lyapunov for forecast steps $N_\mathrm{f}$ ranging from 1 to 5 for the reference experiment. It is clear that the choice of the smallest value for $N_\mathrm{f}$ yields better results for both metrics. We argue however that this behaviour is specific to the inherent chaotic properties of the L96 model. For instance, a lead time as high $N_\mathrm{f} = 5$ makes

5    the neural network minimizes its loss at a remote lead time when the state has become too unpredictable (half of the Lyapunov time). The neural network is then too sensitive to the "butterfly effect" and hampers the convergence of the training step.

### 4.5.2 Model error

Figure 10 shows the sensitivity of the same skills scores with respect to the model noise specified in the DA process in Eq. (6). Regarding the forecast skills, the optimal value seems to be $\sigma^\mathrm{m} \approx 1.0$ whereas the reconstruction of the long-term dynamic favours a much smaller $\sigma^\mathrm{m} \approx 0.01$. It means that this parameter can be adjusted to the purpose of the surrogate model: either yielding accurate forecast or reconstructing consistently the long-term dynamics. In our case, we have selected a compromise by setting $\sigma^\mathrm{m} = 0.1$.

We tentatively explain the reason for this trade-off between forecasting skill and consistent reconstruction as follows. The larger $\sigma^\mathrm{m}$ is, the larger the spread of the ensemble in the DA step and the more efficient is the DA. On the other hand, the DA updates interfere with the calculation of the Lyapunov exponents. With a small model error $\sigma^\mathrm{m}$, such jumps in the time series are much smaller.

A perspective of this work, which is outside the scope of this paper, would be to propose some methodology to estimate the model error statistics (see e.g. Pulido et al., 2018). Note finally that, up to a certain extent, the implicit estimation of the multiplicative inflation in the EnKF-N can compensate for some model error even if was not initially targeted for that (Raanes et al., 2019).



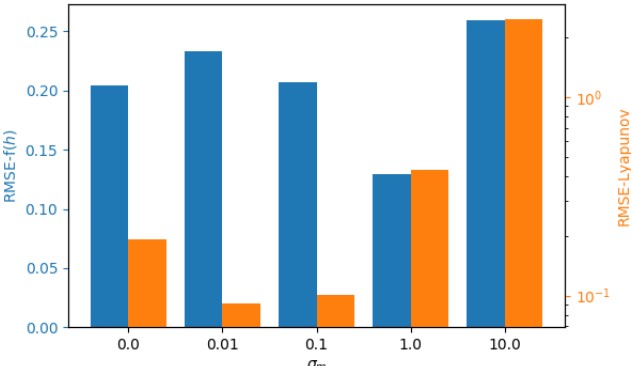

**Figure 10.** RMSE-f (blue) and RMSE-Lyapunov (orange) with respect to the standard deviation of the model error $\sigma^{\mathrm{m}}$, Eq. (6). 50% of the field is observed with a noise $\sigma^{\mathrm{obs}} = 1$.

## 5    Conclusions

We have proposed a methodology to build a data-driven surrogate model from partial and noisy observations. The method is based on an iterative algorithm. At each iteration, a data assimilation step interpolates and de-noises the observations, then alternates with a machine learning step that learns the underlying dynamics of the DA analysis.

The approach has been illustrated on a spatially extended (one dimensional) 40-variable Lorenz chaotic model (Lorenz and Emanuel, 1998) and evaluated by three main skills scores. First, we have evaluated the ability to interpolate and de-noise the observational field. In our reference experiment (50% of the field observed, with a standard deviation of noise set to 1), it has been shown that the proposed method was able to reconstruct the complete field with a smaller error than the observational error. Second, we have evaluated the dynamical features of the surrogate model compared with the true L96 model. We have

shown that the more energetic time scales (frequency lower than 5 Hz corresponding to $t > 0.33$ in Lyapunov time unit), and the positive Lyapunov exponents are well represented by the surrogate model. Finally, we have evaluated the forecast skills of the model, achieving accurate forecasts out to 2 Lyapunov time units (twice the typical "memory" of the system).

Sensitivity experiments to the level of observation noise and to the density of available observations have been conducted. The method is very sensitive to the level of noise, as expected, and the forecast skill gets worse with an increasing noise level.

If less than 50% of the field is observed, the results also deteriorate but as long as the fraction of observations exceeds 50% there is no significant difference in the skills of the surrogate model. In the case of a fully observed field without noise, we reproduce the skills found in similar machine learning configurations although not as accurate as in Bocquet et al. (2019) in the case of an identifiable model.

It has also been shown that by tuning certain parameters of the algorithm (number of forecast steps of the neural network

and standard deviation of the model noise in data assimilation), it was possible to favour the forecast skill over the long-term dynamics reconstruction or vice-versa.



One drawback of the method is the computation cost. The algorithm needs to apply one complete data assimilation procedure (equivalent to 30 forward model runs to propagate the ensemble members) and one neural network training (equivalent to 20 forward model runs) for each cycle until convergence. In this work, we have not optimized the computational cost of the algorithm, but there are several avenues for improvement of that aspect. We could for example set an optimal stopping criterion,

or leverage from parallel computing (the neural network could start training before the end of the data assimilation, using the portion of the assimilation run which is already analysed).

Arguably, the success of the method relies at least partially on the autonomous character of the L96 model. This makes sure that learning from past analogues is valuable for forecasting purposes. Nevertheless, we foresee that the approach can be further extended to non-autonomous large scale systems by considering that the system variations are slow or by including

the external forcing as an input of the neural network. A major feature of the proposed algorithm is its extreme flexibility. It is possible to plug any data assimilation scheme and machine learning schemes independently. For example, from an existing data assimilation system, it could be feasible, without much change in the existing system, to apply this algorithm to estimate the surrogate model of a non-represented dynamical process in the original numerical model (bias, parameter, ...) or to replace one particular heuristic sub-model by a trained neural network. The latter would result in a hybrid numerical model, including

trained neural networks, that honours the fundamental equations of the motions and only relies on machine learning for a data-driven sub-model. In term, one could envision that model tuning and data assimilation could be carried out in the same framework.

*Code availability.*   The code used for the experiments presented in this paper is available on https://github.com/brajard/GMD-code release 1.1

*Author contributions.*   JB first proposed the theory, implemented and conducted the numerical experiments. All authors have contributed to the interpretation of the theory and of the results as well as the edition of the manuscript. The authors approved the manuscript for publication.

*Competing interests.*   The authors declare they have no conflict of interest

*Acknowledgements.*   JB, AC and LB have been funded by the project REDDA (#250711) of the Norwegian Research Council. CEREA and LOCEAN are members of Institut Pierre–Simon Laplace (IPSL).



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
