# Peer review of "Combining data assimilation and machine learning to emulate a dynamical model from sparse and noisy observations: a case study with the Lorenz 96 model"

_Geoscientific Model Development, 2019_

## Referee Comment (RC1) · Peter Düben (Referee) · 6 Jul 2019

This is very good and timely paper that should be accepted in GMD. The combination between deep learning and data assimilation as studied in this paper has great potential and the paper could basically be published as it is. However, the authors may decide to adjust it slightly following the minor comments below.

- Section 3.3.2: This is quite a specific network architecture that you are using. Can you provide more detail how you discovered it? Can you speculate whether

[Figure]

the "x" and "+" are required due to the underlying shape of the equations of the Lorenz model?

- I assume that you have 1-D periodic boundary conditions for the network.

- Figure 2: It took me a while to understand that 2a, 2b and 2c are in parallel. This is not intuitive from the figure. However, I am not sure how to improve this.

- What would happen if some of the parameters would never be observed during the training period?

- Figure 4: This may be my ignorance but I would have expected to see the high frequencies to be correct and the low frequencies to be incorrect since you are basically training on a timestep level. Do you have any comments on this?

- Section 4.5.2: Could this configuration therefore be used to tune stochastic parametrisation schemes? This could maybe be discussed. We had some success using GANs and dropout methods to develop neural network parametrisation schemes for Lorenz 95 that showed some variability.

- P18: Could this also be made more efficient by training on interpolated observations in a first instance with no need to use the entire data assimilation scheme? Once the neural network model has converged here, the data assimilation configuration could be use to refine it.?

- Caption Figure 7. "0. 50

- Caption Figure 8: "with with"

- P16: "if was" -> "if it was"

- P18: "parallel computing" I would suggest to call this "concurrent computing" since you do not refer to standard MPI/OpenMP parallelisation.

- "resolvent" could be explained a bit more.

- Abstract: "applies alternatively" could be re-phrased.

- P2 l11: "precipitations" -> "precipitation"

- P5 l1 and P7 l14: There are unnecessary line breaks.

- P7 l3: It could be stated that $\sigma^{obs}$ is a rather arbitrary choice at this stage.

- One of the references is incomplete: "E, W.:"

---

## Referee Comment (RC2) · Anonymous Referee #2 · 12 Jul 2019

Review of the paper
« Combining data assimilation and machine learning to emulate a
dynamical model from sparse and noisy observations: a case study
with the Lorenz 96 model»

**General comments**

This paper introduces an algorithm to learn a surrogate model based on neural networks, from noisy and partial observations. The proposed algorithm is tested on the Lorenz 96 model using simulations.

This is an interesting topic but the work seems incomplete in both the methodology and its evaluation.

**Specific comments**

Many choices in the algorithm seem arbitrary and should be further discussed and validated through simulations. In particular, the authors state on Page 6 "The procedure can be viewed as an expectation-maximization algorithm...". I believe that it is a good idea to use the EM machinery but I see fundamental differences between the proposed algorithm and the EM algorithm. I think that these differences may deteriorate significantly the performances of the proposed algorithm. These differences should be highlighted, discussed and their impact precisely validated through simulations. Some of these differences are highlighted below.

1. Page 4, l 25-30. "Our choice of the EnKF-N is motivated by its efficiency, its high accuracy for low-dimensional systems, and its implicit estimation of the inflation that would otherwise have had to be tuned." In the EM algorithm, the surrogate model is fitted by maximizing a likelihood function based on the smoothing distribution. Here the authors propose to use the filtering (instead of smoothing) distribution and only the conditional expectations of the filtering distribution (instead of the full conditional distribution of $x(k+1)$ given $x(k)$ and the observation $y(1:K)$). The authors motivate this choice by the fact that a smoother "is less common in the operational DA community". I agree that using the proposed approach leads to substantial simplifications, but I also feel that it may deteriorate substantially the estimate of the surrogate model. This should be assessed using numerical simulations to compare the results obtained with the proposed approach with the ones obtained with a smoother.
2. Page 4, eq. (5). Why using the norm associated to $P_k$ here? In the M-step of the EM algorithm, the function to maximize is defined as the expectation of the full log-likelihood function with respect to the smoothing distribution. What is the link with the cost function proposed here?
3. Page 16. "A perspective of this work, which is outside the scope of this paper, would be to propose some methodology to estimate the model error statistics...". I don't agree that it is outside the scope since it may lead to substantial improvements of the proposed approach. The choice of the model error is really ad-hoc, with a diagonal matrix (although I would expect some correlations between the errors on the different components) considered as an hyperparameter with arbitrary value. I believe that the estimation of the model error should be included in the methodology before publication. The estimation of the error covariances matrices with the EM algorithm is discussed for example in Dreano et al.

I also have concerns about the numerical experiments, see below.

1. "The model is integrated over 40,000 time steps (K = 40,000)". It seems to be a huge learning sequence, unrealistic for application in DA! Please vary this value and discuss the sensitivity of the results with respect to K.
2. Figure 8, panel a. In my opinion, this is the more interesting plot. It permits a comparison between the NN model fitted on a "perfect" sequence of the true state without observational error (best surrogate of the true model based on NN) and the NN model fitted on noisy observation, which comes out the proposed algorithm. I have the feeling that a good algorithm should permit to retrieve similar NN model from perfect or noisy observations (this is what you expect to get with the EM algorithm) and this plot suggests that there are important differences between the two NN models, despite the huge learning sequence, and thus that the algorithm should be improved.
3. Section 3.2. Various scores are proposed to measure the "distance" between the true and the surrogate model. If I understand correctly, the two last criteria (Lyapunov spectrum and Power spectrum) are used to measure some long-term statistical properties of sequences simulated with the surrogate model. I think that these criteria are more indirect validation criteria of the fitted model and it should come later in the discussion. I also wonder if other criteria linked to the distribution of the stationary distribution (e.g. mean, covariance) may provide additional information and be easier to interpret.

Page 18. "One drawback of the method is the computation cost". I think that this is a really important point which should be more discussed since one the argument of using ML tools is to reduce computational costs compared to running a more physical model. The authors should explain why it is so costly and give a more precise idea of the computational costs for the numerical experiments done in the paper.

**Technical corrections**

1. Writing: the authors often use "we...". Please avoid.
2. Page 4, eq. (5) and Section 4.5.1. If the idea is to mimic the EM algorithm, the choice Nf=1 is natural since it arises when writing the full likelihood function of a state space model using usual Markovian assumptions. And simulation results in Section 4.5.1 suggest indeed that it is the best choice. Why making things more complicated by introduction this "hyperparameter"? The authors may consider only the case Nf=1.
3. Figure 7, panel a. The long-term forecasts are improved when the observation noise in increased: any idea to explain this?
4. Page 5, l 8-10. "Convolutive layers apply a convolution acting locally around each grid point of the field. It is equivalent to a locality hypothesis, assuming that there are no long-range correlations between the state variables. Note that it does not discard further distance correlation arising from the time integration". Many terms have not been introduced before like "grid point", "field", "locality". The authors may introduce the context before. The last sentence is completely mysterious for me.
5. End of Page 7, "Note however that localization...". Is it really necessary to talk about localization here? If yes, please explain what it means.
6. Title of Section 4.1 "Convergence of the algorithm". I do not see any convergence here, only some criteria which decrease.
7. Page 11, l3. "The former is the RMSE of a field obtained via quadratic interpolation

without any use of a dynamical model (which is instead essential in DA)". You could also compare with a space-time interpolation without the model to be fair.

8. Some references are incorrect (e.g. the paper by Weinan).
9. Lots of papers cited in the references were written by the authors of the present paper, and, sometimes, I have the feeling that it is a bit artificial.

---

## Author Comment (AC1) · 23 Jul 2019

Dear reviewer,

We wish to sincerely thank the Reviewer for his/her insightful revision of our manuscript. We have no doubts that, addressing most of the points in there, will result in an improved version of our work. We look forward to work on producing a full response and a revised version.

Nevertheless, before doing that, it urges us to clarify what, in our opinion, appears is

a key misunderstanding on the main goals and objectives of our work. This misunderstanding seems to be the driver of the Reviewer main concerns and therefore we consider important to clarify this as soon as possible. These are the two main controversial points.

1) EM algorithm.

The reviewer makes a lot of meaningful remarks on the EM algorithm. These remarks will be very helpful in the future. But the reference in the paper to the expectation-maximization algorithm was very general and qualitative and was made in the purpose of contributing to the general reflexion on this type of methods. We don't claim that our method is an EM algorithm, and in fact, for the reasons pointed out by the reviewer, it is not. We are trying to minimize the classical regression cost used in machine learning, but, as the state of the system is unknown, we are feeding the neural network by estimation of this state using data assimilation. EM method was never within the design nor the conception of our approach.

2) Model error.

We are totally in line with the reviewer in that model error is an important point in general, and a possible improvement of our method in particular. It is also true that choices have to be made in the practical implementation of our algorithm. As such, we have made a choice of a diagonal model error which is standard for the toy models employed and presents a lot of advantages in term of cost and stability. This particular choice (among other: e.g.the architecture of the neural network, the choice of ensemble data assimilation) is already leading to good performances. The fact that there is still room for improvement is inherent to all implementation and shows, in our opinion, the high potentiality of this method.
* * *

---

## Short Comment (SC1) · 26 Jul 2019

Thank you for putting your code online. There is just one technical issue with this which needs to be fixed in the revised submission. GitHub is an excellent development and distribution platform, but it is not a citable archival location suitable for scientific publications. Even GitHub themselves tell you to use Zenodo for this! Please therefore follow the instructions at https://guides.github.com/activities/citable-code/ to produce a citable archive. The resulting archive page will even have a download button for importing the citation into BibTeX (or many other reference managers).

[Figure]

For full details on the GMD code and data availability policy, please see: https://www.geoscientific-model-development.net/about/code_and_data_policy.html.

---

## Author Comment (AC2) · 26 Aug 2019

Dear Peter Düben,

we first wish to thank you for your comments and suggestions that have helped us to improve the manuscript. We report below your original remarks (in blue) followed by our responses.

*Section 3.3.2: This is quite a specific network architecture that you are using. Can you*

[Figure]

*provide more detail how you discovered it? Can you speculate whether the "×" and "+" are required due to the underlying shape of the equations of the Lorenz model?*

The following explanation was added to section 3.3.2: "The specific bilinear layer aims at facilitating the training in the case where multiplications are involved in the true model. This design is therefore based on a priori knowledge of the underlying model, given that multiplicative terms are ubiquitous in ODE-based geophysical models. For instance, this is the case of the model described in Eq. (9) used to illustrate the work."

*I assume that you have 1-D periodic boundary conditions for the network*

You are right, this point was explicitly stated in section 3.3.2: "As the numerical experiments in the following are conducted on a periodic spatial domain, (see section 3.1), the input of the neural network is accordingly 1-D periodic at the boundaries."

*Figure 2: It took me a while to understand that 2a, 2b and 2c are in parallel. This is not intuitive from the figure.*

It is true the figure 2 was unclear, a new figure (hopefully more intuitive) is proposed in the new version of the paper (See figure 1)

The situation of unobserved portions of the space would be more challenging. This aspect has now been clarified in section 3.1: "Note that variables have a non-zero probability to be observed in some instance. In the more challenging case where portions of the field are never observed (e.g. fixed observing network, hidden parameter, ...), the setup may have to be adapted to account for an unobserved latent subspace, as it proposed for instance in Ayed et al. (2019)."

*Figure 4: This may be my ignorance but I would have expected to see the high fre-*

*quencies to be correct and the low frequencies to be incorrect since you are basically training on a timestep level. Do you have any comments on this?*

We have elaborated on this result in section 4.3 by adding the following comment: "The errors are thus higher at higher frequencies and there is no significant improvement throughout the optimization process (see the grey and orange lines after 5 Hz in Fig. 4). The low frequencies are better observed (there are 4 times as many observations for each 2 Hz oscillation as for an 8 Hz oscillation) and better reproduced after the DA step, whereas high-frequencies are not reproduced from observations."

*Section 4.5.2: Could this configuration therefore be used to tune stochastic parametrisation schemes? This could maybe be discussed. We had some success using GANs and dropout methods to develop neural network parametrisation schemes for Lorenz 95 that showed some variability.*

We agree it is actually an ongoing direction of extension of the current work we are considering. The following sentence was added at the end of section 4.5.2: "Another ongoing direction of the extension of the current work could be to estimate stochastic model errors. For example generative models (generative adversarial networks or variational auto-encoders) could be used to tune stochastic parametrisation schemes."

*P18: Could this also be made more efficient by training on interpolated observations in a first instance with no need to use the entire data assimilation scheme?*

Actually, this is what is done (see section 3.3.3). But thank you for stressing out that this point needs clarification. Following your comment, we also stress the gain in term of computing time by adding the following sentence in section 3.3.3: "Starting with random weights as it is usually done in ML training algorithms could lead to convergence problem of the two-step algorithm and also will be inefficient in term of computing time."

*- Caption Figure 7. "0. 50*
*- Caption Figure 8: "with with"*
*- P16: "if was" -> "if it was"*
*- P18: "parallel computing" I would suggest to call this "concurrent computing"*
*since you do not refer to standard MPI/OpenMP parallelisation.*
*- "resolvent" could be explained a bit more.*
*- Abstract: "applies alternatively" could be re-phrased.*
*- P2 l11: "precipitations" -> "precipitation"*
*- P5 l1 and P7 l14: There are unnecessary line breaks.*
*- P7 l3: It could be stated that $\sigma$*
*obs is a rather arbitrary choice at this stage.*
*- One of the references is incomplete: "E, W.:"*

Thanks for pointing out those points. They have been addressed in the new version of the paper. Except for the line break P7 l14 which was intended to make the equation compatible with a two columns format. Also, the E.W. reference is correct. Although unusually short, E is the real family name of the first author.

[Figure]

**Fig. 1.** Proposed neural network architecture for the surrogate model. The input layer is to the left.

---

## Author Comment (AC3) · 26 Aug 2019

Dear reviewer,

we first wish to thank you for your comments and suggestions that have helped us to improve the manuscript. We have already provided a first response to the main comments you made. We report below your original remarks (in blue) followed by our responses.

*Many choices in the algorithm seem arbitrary and should be further discussed and validated through simulations. In particular, the authors state on Page 6 "The procedure can be viewed as an expectation-maximization algorithm...". I believe that it is a good idea to use the EM machinery but I see fundamental differences between the proposed algorithm and the EM algorithm. I think that these differences may deteriorate significantly the performances of the proposed algorithm. These differences should be highlighted, discussed and their impact precisely validated through simulations. Some of these differences are highlighted below.*

Thank you very much to point that out. As it was mentioned in our previous answer, we think that the main objective of our work was unclear. The resemblances with the EM algorithm was meant as an analogy rather than a strict mathematical equivalence. The EM algorithm has never been in the design of our approach. We apologize if this reference has raised false expectations. As such, the misleading reference to the EM algorithm was removed in the corrected version of the paper.

*1. Page 4, l 25-30. "Our choice of the EnKF-N is motivated by its efficiency, its high accuracy for low-dimensional systems, and its implicit estimation of the inflation that would otherwise have had to be tuned." In the EM algorithm, the surrogate model is fitted by maximizing a likelihood function based on the smoothing distribution. Here the authors propose to use the filtering (instead of smoothing) distribution and only the conditional expectations of the filtering distribution (instead of the full conditional distribution of x(k+1) given x(k) and the observation y(1:K)). The authors motivate this choice by the fact that a smoother "is less common in the operational DA community". I agree that using the proposed approach leads to substantial simplifications, but I also feel that it may deteriorate substantially the estimate of the surrogate model. This should be assessed using numerical simulations to compare the results obtained with the proposed approach with the ones obtained with a smoother*

Let us remind first that we are not in an EM formalism. Thanks to your comment, we

have nevertheless added the following sentence in section 2.2: "This time-dependent matrix aims at giving weights to the states during the optimization process depending on how uncertain they are: an uncertain state will have a low weight."

We understand here that you suggest using, instead of the diagonal matrix $P_k$, the full covariance matrix that could take into account observations errors, co-variances and temporal auto-correlation. One possible advantage to use a diagonal $P_k$ is the algorithmic simplicity in high dimensions: The diagonal of $P_k$ is practically the only accessible quantity that would not require enormous storage space. The effect of using a more complete $P_k$ (which would also likely be more costly) is an interesting perspective of this work but considered as an already challenging task in data assimilation, i.e. even when the model is relatively well known.

*3. Page 16. "A perspective of this work, which is outside the scope of this paper, would be to propose some methodology to estimate the model error statistics...". I don't agree that it is outside the scope since it may lead to substantial improvements of the proposed approach. The choice of the model error is really ad-hoc, with a diagonal matrix (although I would expect some correlations between the errors on the different components) considered as an hyperparameter with arbitrary value. I believe that the estimation of the model error should be included in the methodology before publication. The estimation of the error covariances matrices with the EM algorithm is discussed for example in Dreano et al.*

Here again, we thank you to acknowledge the importance of model error estimation in data assimilation. We have started to give an answer to your comment in our first response. We totally agree, as we have mentioned in the paper, that a more complex estimation of the model error is a natural perspective of the work we propose. Nevertheless, we still think that it is outside the scope of the present paper. The automatic estimation of model errors would bring an extra level of complication to the algorithm, which will make the paper less easily accessible. It is the fate of every

algorithm to grow gradually more complex over consecutive studies, but documenting the performance of an algorithm in its simplest form is also a valuable starting point. Note that Dreano et al. 2017 only applied their methodology to the 3-variable Lorenz 63 model. There is already a big complexity gap between the L63 and L96 model, the former one being considered 0-dimensional and the latter one 1-dimensional.

*I also have concerns about the numerical experiments, see below. 1. "The model is integrated over 40,000 time steps (K = 40,000)". It seems to be a huge learning sequence, unrealistic for application in DA! Please vary this value and discuss the sensitivity of the results with respect to K.*

Thank you for pointing out that critical point. As it is stated in the introduction, the general assumption of this work is to assume that we can benefit from a huge number of data (particularly satellite data). But we fully agree that the number of needed data to estimate both the underlying dynamics and the state of the system is a question that needs to be addressed. We have conducted a sensitivity experiment to the size of the training set. We have added this figure (Fig. 1) . We also added the following comment to the paper: "Figure 1 shows the value of RMSE-f and RMSE-Lyapunov for three different training test size $K \in \{4 \times 10^2, 4 \times 10^3, 4 \times 10^4\}$. Note that $K = 4 \times 10^4$ is the value used for all the other experiments in this paper. As expected, the error of the algorithm for both metrics decreases slightly as the size of the training set increases. Note that the amplitude of the scale used in this figure is small compared with the other sensitivity experiments. The result shows that it is possible to produce a fair surrogate model with this algorithm given smaller datasets.

It is worth noticing that making the algorithm converge is more challenging with $K < 4 \times 10^4$. We interpret this behaviour by the fact that the neural network can overfit the dataset during the training phase. In this particular experiment, the number of epochs for each cycle has been reduced to 5 (instead of 20 in the other experiments as stated in table 1) to mitigate overfitting. It also very likely

that this result depends on the underlying dynamics to be retrieved and that another dynamics than the L96 model could show different sensitivity to the size of the dataset."

*2. Figure 8, panel a. In my opinion, this is the more interesting plot. It permits a comparison between the NN model fitted on a "perfect" sequence of the true state without observational error (best surrogate of the true model based on NN) and the NN model fitted on noisy observation, which comes out the proposed algorithm. I have the feeling that a good algorithm should permit to retrieve similar NN model from perfect or noisy observations (this is what you expect to get with the EM algorithm) and this plot suggests that there are important differences between the two NN models, despite the huge learning sequence, and thus that the algorithm should be improved.*

This is true that a good algorithm is able to retrieve very close results with and without noise, but the extent of this in practice is challenging. By consequence, a discussion was added to the paper in section 4.4: " If 100% of the field is observed with an error corresponding to $\sigma^{\mathrm{obs}} = 1$, there is still a significant degradation with respect to the perfect case $\sigma^{\mathrm{obs}} = 0$. This can be due to several factors: first, there is a severe loss of information when noise is added; then, the DA step (which is not necessary in the perfect case) does not provide a perfect estimator of the state (even in the case of a perfect model); finally, the algorithm itself is also a source of error as there it could converge toward a local minimum."

Besides that, you suggest that the EM algorithm should be almost totally unaffected by noise or by missing data, given a long enough training sequence. Beyond trivial linear systems, we are not aware of any literature to support this claim and we would truly appreciate additional references from you on that matter.

*3. Section 3.2. Various scores are proposed to measure the "distance" between the true and the surrogate model. If I understand correctly, the two last criteria (Lya-*

*punov spectrum and Power spectrum) are used to measure some long-term statistical properties of sequences simulated with the surrogate model. I think that these criteria are more indirect validation criteria of the fitted model and it should come later in the discussion. I also wonder if other criteria linked to the distribution of the stationary distribution (e.g. mean, covariance) may provide additional information and be easier to interpret.*

Following your feedback, we have better justified the choice of scores in the beginning of section 3.2: "The results of this work is not focused on a specific application (e.g. forecasting, climate simulation), thus several scores are calculated in order to validate the model regarding various properties: interpolation, forecast and long-term dynamics. In the context of a specific application, it would be possible to focus specifically on one aspect."

We have also added the mean of the field as suggested. The definition of the average was added in Eq.(12) and the following sentence was added in section 4.3.: "The average value of the true model is 2.35 whereas the one of the surrogate model is 2.30. The surrogate model has a small negative bias due to the underestimation of positive extreme values." We have attached to this answer the QQ-plot (see Fig 2) of the surrogate model values compared with the true model values to demonstrate the underestimation of positive extreme values.

Regarding the variance, as the PSD is the Fourier transform of the auto-correlation, it appears to be redundant with the variance.

*Page 18. "One drawback of the method is the computation cost". I think that this is a really important point which should be more discussed since one the argument of using ML tools is to reduce computational costs compared to running a more physical model. The authors should explain why it is so costly and give a more precise idea of the computational costs for the numerical experiments done in the paper.*

You are right that it is a common claim that ML tools reduce computational costs. In our case, we are not focusing on the computational aspect of ML but on the ability of a neural network to emulate a dynamical model from observations. The discussion on the computational cost in the conclusion has been a bit extended by referring to the possibilities of deep-learning libraries you have mentioned.

*1. Writing: the authors often use "we...". Please avoid.*

Thank you for this comment. Indeed, the word "we" was sometimes used unspecifically. We have removed the instance of "we" in the case where it does not refer specifically to the authors of the paper.

*2. Page 4, eq. (5) and Section 4.5.1. If the idea is to mimic the EM algorithm, the choice Nf=1 is natural since it arises when writing the full likelihood function of a state space model using usual Markovian assumptions. And simulation results in Section 4.5.1 suggest indeed that it is the best choice. Why making things more complicated by introduction this "hyperparameter"? The authors may consider only the case Nf=1.*

Now we have tried to make clear that we were not trying to mimic the EM algorithm, it is still fair to assess this hyperparameter. We have kept this hyperparameter because it appears that in some specific case (e.g. when the neural network is trained on interpolated data), it gives better result in term of cross-validation to use $N_f > 1$.

*Figure 7, panel a. The long-term forecasts are improved when the observation noise in increased: any idea to explain this?*

The value of RMSE-f after the predictability horizon cannot be interpreted as an indicator of improvement. After this horizon ($i$ large enough), it is fair to assume that $\mathcal{G}_{L96}^{(i)}(\mathbf{x}_0^p)$ and $\mathcal{G}_{\mathbf{W}}^{(i)}(\mathbf{x}_0^p)$ are uncorrelated, and as a consequence (in a simplified

notation), RMSE-f = Var(True Model) + Var(surrogate model). As the variance of the true model does not change in the experiment in Fig. 7, the decreasing of RMSE-f is due to the decreasing of the variance of the surrogate model (asymptotically) as it is stated in section 4.4. We have slightly modified the text to make it clearer.

*4. Page 5, l 8-10. "Convolutive layers apply a convolution acting locally around each grid point of the field. It is equivalent to a locality hypothesis, assuming that there are no long-range correlations between the state variables. Note that it does not discard further distance correlation arising from the time integration". Many terms have not been introduced before like "grid point", "field", "locality". The authors may introduce the context before. The last sentence is completely mysterious for me.*

We have rephrased this part to better introduced the notation and make this important claim clearer: "Convolutive layers apply a convolution acting in a local spatial neigbourhood around each state variable $x_{n,k}$ of the field $\mathbf{x}_k = [x_{0,k}, \cdots, x_{m-1,k}]$. It is equivalent to a locality hypothesis, assuming that there are no long-range correlations between the state variables. The locality hypothesis is used in ensemble DA in the localization process to discard spurious correlation due to the limited number of members in the ensemble (Evensen, 2003). Note that it does not discard further distance correlation arising from the time integration."

*5. End of Page 7, "Note however that localization...". Is it really necessary to talk about localization here? If yes, please explain what it means.*

Now the localization was introduced in section 2.2, we hope that this this reference is clearer.

*6. Title of Section 4.1 "Convergence of the algorithm". I do not see any convergence here, only some criteria which decrease.*

The term "convergence" was ambiguous in this context. We have clarified it by specifying in the text that "the proposed algorithm is converging toward a stable value".

*7. Page 11, l3. "The former is the RMSE of a field obtained via quadratic interpolation without any use of a dynamical model (which is instead essential in DA)". You could also compare with a space-time interpolation without the model to be fair.*

Our formulation was misleading. We are actually performing a space-time interpolation using the "cubic" method of the scipy function `griddata`. The space-time interpolation has been specified in the text and the term "quadratic" has been replaced by "cubic".

*8. Some references are incorrect (e.g. the paper by Weinan).*

We have reviewed the reference list, and make an update. Also, the E.W. reference is correct. Although unusually short, E is the real family name of the first author.

*Lots of papers cited in the references were written by the authors of the present paper, and, sometimes, I have the feeling that it is a bit artificial.*

We have always tried to give the more adequate citation. Nevertheless, if you provide the places where you find the citations are artificial or where there would be a more relevant reference, we are totally ready to modify the citations.

[Figure]

**Fig. 1.** RMSE-f (blue) and RMSE-Lyapunov (orange) with respect to the size of the training set $K$. 50\% of the field was observed with a noise $\sigma^\textrm{obs}=1$.

[Figure]

**Fig. 2.** Quantile-quantile plot of the surrogate model values compared with the true model values

---

## Author Comment (AC4) · 26 Aug 2019

Dear editor,

we first wish to thank you for your comment. The zenodo deposit available on http://doi.org/10.5281/zenodo.2925547 was already indicated in the asset section of the paper, but we also added the reference to the code doi in the code availability section of the paper.

---

## Author Comment (AC5) · 28 Aug 2019

There was an error on the subject of the previous message which the "answer to reviewer 1" (and not reviewer 2 as it is wrongly stated).

with our apologies.